# The impact of digital transformation on open innovation performance: The intermediary role of digital innovation dynamic capability

**Peiyao Qiu** [1]*, **Benrui Chang** [2]

1 Henan University of Technology, School of Economics and Trade, Zhengzhou, Peoples R China,
2 Tianjin University, Economics School, Tianjin, Peoples R China

* 407853679@qq.com

## Abstract

In the context of digital economy, both the process and boundaries of open innovation performance are changing, and digital transformation has become an important factor for enterprises to realize open innovation performance in a complex and changing environment. This study based on the dynamic capability theory, etc. to explore how digital transformation drives the open innovation performance of enterprises and analyze its impact effects as well as the internal mechanism of action. In this paper, A-share listed companies in China's Shanghai and Shenzhen cities from 2008–2022 are selected as research samples for empirical tests, and the research results show that digital transformation significantly stimulates enterprises' open innovation performance, which still holds true after relying on a series of robustness tests. The study on the mechanism of action shows that digital transformation mainly enhances the dynamic capability of digital innovation, and ultimately promotes the open innovation performance of enterprises. And environmental dynamics play a positive moderating role. Heterogeneity analysis shows that the incentive effect of digital transformation on open innovation performance is more obvious in enterprises with high-tech and non-state ownership. Based on the new perspective of digital innovation dynamic capability, this study further reveals the effect and mechanism of digital transformation on open innovation performance, and provides theoretical basis and decision-making reference for enterprises to utilize the opportunity of digitalization to achieve open innovation performance results.

## 1. Introduction

The digital economy accelerates innovation breakthroughs in key core technologies and important products of enterprises. The massive emergence of digital elements and the rapid iteration of digital technology bring new windows of opportunity for enterprises' open innovation performance, but also make organizational innovation activities face great challenges. Gradually blurred and open innovation boundaries continue to give rise to new organizational forms, digital elements with mobility and sharing weaken the exclusivity of traditional innovation resources [1], and traditional innovation capabilities are difficult to ensure that organizations can effectively identify internal and external open innovation

**Data availability statement:** All relevant data are within the manuscript and its Supporting information files.

**Funding:** The authors received no specific funding for this work.

**Competing interests:** The authors have declared that no competing interests exist.

performance opportunities. In order to ensure the efficient implementation of activities such as aggregating and reorganizing innovation factors, innovation capabilities urgently need to be digitally upgraded. Constructing digital innovation capabilities through digital transformation is a necessary means for organizations to cope with the challenges of digital change and digital resources in dynamic innovation environments [2]. Therefore, by clarifying the direction of digital upgrading of dynamic innovation capabilities, and revealing the impact of digital transformation on the open innovation performance of enterprises in highly dynamic environments, it is important for enterprises to seize the opportunities of digital technological change to achieve key technological Therefore, it is of great significance to reveal the impact of digital transformation on the open innovation performance of enterprises in a highly dynamic environment by clarifying the direction of digital upgrading of dynamic innovation capability, so as to realize key technological breakthroughs to establish new innovation competitive advantages by seizing the opportunity of digital technology change.

Digital transformation provides a feasible path to empower corporate innovation [3]. With the rapid development of the digital economy, digital technologies such as big data, cloud computing, and the internet of things have broken the traditional innovation boundaries and effectively constructed a three-dimensional innovation network of firms, consumers, and online networks in the era of the digital economy with no spatial and temporal constraints [4]. These technologies have greatly influenced the communication pathways between innovators and consumers, and shifts in consumer demand have accelerated the integration of corporate innovation practices with digital technologies, resulting in consumer-focused digital innovation becoming a new tool for companies to meet market challenges and upgrade their innovations. According to the 51st Statistical Report on China's Internet Development, as of December 2022, China's Internet penetration rate reached 75.6%, the average weekly Internet usage time per capita was 26.7 hours, the number of Internet users was 1.067 billion, and the number of online shopping users reached 845 million, accounting for 79.2% of the overall number of Internet users, which lays a solid foundation for the development of enterprise digital innovation. At the same time, compared with traditional enterprises, new-age enterprises have relatively flat organizational structure and flexible decision-making, with lower decision-making inertia and resource rigidity, and prefer digital innovation that is more accessible to consumers.

With the rapid development of digital technology, more and more scholars have begun to pay attention to the emerging concept of digital transformation. Currently, fewer studies have explored the impact of digital transformation on firms' open innovation performance. Some studies have conducted extensive research on the intricate relationship between digital transformation and corporate innovation from the perspective of digital empowerment or information paradox [5]. Some scholars argue that different from traditional innovation [6], digital transformation can digitally transform and upgrade the content and methods of traditional innovation through digital technology with more depth and breadth, which breaks through the limitations of innovation resources of traditional innovation with diversified channels, precise targeting, and efficient communication [7] and integrates different resources with the help of digital media, thus realizing the innovative resources that traditional innovation has been more difficult to achieve. By integrating different resources through digital media, we can realize the value co-creation that is difficult to reach by traditional innovation, which is more conducive to the "post-Schumpeterian" innovation paradigm of enterprises focusing on market users [8], and thus improve the efficiency of the transformation of innovation results. At the same time, some scholars believe that digital transformation uses digital technology as a technological base [9], pays more attention to changes in the core business, strategy and innovation model of the enterprise, and reconstructs the enterprise value creation system to

promote the development of enterprise innovation. It can be seen that the literature has begun to study the relationship between digital transformation and enterprise innovation. Moreover, the current research on the impact mechanism of digital transformation on enterprises' open innovation performance has gradually attracted the attention of scholars. Some scholars discuss how digital transformation can promote enterprises' innovation performance by affecting knowledge management ability from the perspective of knowledge management, and believe that digital transformation plays a key role in knowledge acquisition, flow and reorganization within enterprises [10,11]. In addition, digital programmability means that knowledge management methods have been innovated [12]. Some other scholars analyze the impact mechanism from the perspective of enterprise process digitalization, and believe that digital transformation can affect the product design, supply chain management and marketing of enterprises, and will promote enterprises to further break through organizational inertia and improve their innovation performance [13,14]. While, the current research on the mechanism of digital transformation affecting enterprise open innovation involves few angles and is relatively concentrated. Therefore, other perspectives need to be expanded to enrich the relevant research.

However, as can be seen from the above analysis, previous studies have less explored the impact of digital transformation on enterprise open innovation performance, and less explored the impact mechanism of digital transformation on enterprise open innovation performance from the perspective of digital innovation dynamic capabilities. Digital innovation dynamic capability is an emerging concept proposed by academics in the face of digital upgrading of organizational innovation capabilities in digital innovation contexts [15]. The improvement of digital innovation dynamic capabilities can help companies to collect, store and analyze large amounts of data. Through modern technologies such as data analytics, firms become more agile in sensing new innovation opportunities and future innovation trends. With the help of digital innovation dynamic capabilities enterprises have a more accurate control of market demand, on the basis of which they can make innovative decisions and designs. For example, enterprises can apply cutting-edge technologies such as artificial intelligence, machine learning, and the Internet of Things to improve product design, optimize production processes, and provide personalized user experiences, etc. Enterprises can implement innovations more quickly and iterate rapidly. Therefore, it is worth exploring in greater depth how digital transformation can improve the dynamic capabilities of digital innovation and thus promote open innovation performance in enterprises.

Therefore, this paper explores the impact effect of digital transformation on the open innovation performance of enterprises based on existing research, and explores the role mechanism of digital transformation on the open innovation performance of enterprises from the perspective of dynamic capability of digital innovation, and further explores the moderating role played by the dynamic nature of technology. This paper is committed to correctly analyzing the multidimensional impact of digital transformation on enterprise open innovation performance from the micro new perspective of combining digital innovation and dynamic capabilities, and providing new explanations and evidence on how to achieve enterprise innovation capability improvement in a dynamic environment. This paper tries to answer the following questions:

(1) How does digital transformation affect open innovation performance? Does it promote or hinder innovation?

(2) How does digital innovation dynamic capability mediate the relationship between digital transformation and open innovation performance in firms?

(3) How does environmental dynamics moderate the relationship between digital transformation and open innovation performance in firms?

Compared with the existing literature, the innovation of this paper mainly lies in the following aspects: firstly, it innovatively incorporates digital transformation into the analytical framework of the development of enterprise open innovation performance, and systematically analyzes the effect and mechanism of digital transformation on enterprise open innovation performance. Secondly, it expands the theory of dynamic capability to the field of digital management, introduces the new concept of dynamic capability of digital innovation, and systematically explores the mechanism of digital transformation affecting the open innovation performance of enterprises from the dimensions of digital agility capability and restructuring innovation capability, which clarifies a reliable path for the development of open innovation performance of enterprises in the era of digitalization by effectively relying on digital means, and also broadens the research margins of dynamic capability. It also broadens the research margins on dynamic capabilities. Once again, taking listed companies as samples, we explored the text analysis method to systematically construct a more complete digital transformation measurement index, which makes up for the inadequacy of the existing micro-level enterprise digital transformation metrics, and provides a basis and reference for the subsequent research on digital transformation. Finally, from the aspects of the nature of enterprise property rights and high-tech attributes, we explore whether digital transformation has a heterogeneous impact on the performance of enterprise open innovation, which is conducive to providing heterogeneous practical strategies for the effective promotion of digital transformation and the coordinated development of enterprise open innovation in different scenarios.

## 2. Theoretical background

### 2.1 Dynamic capabilities theory and digital innovation dynamic capability

Dynamic capability refers to an enterprise's ability to integrate, structure and reconfigure internal and external resources to adapt to a rapidly changing external environment. Different from ordinary capabilities that work on a certain production link to accomplish basic tasks such as management and operation, dynamic capabilities are a kind of higher-order capabilities that involve higher-level activities, including sensing, capturing and transforming to sustain the enterprise to guide other capabilities and resources. In the face of the impact of digital technological change on the original enterprise innovation capability theory, a series of digitalization-related capabilities have been developed based on the framework of dynamic capability theory to cope with various digital situations. Under the framework of dynamic capabilities theory, how enterprises perceive and respond to external innovation opportunities, and utilize digital-related innovation resources and capabilities to gain and maintain innovation competitive advantages has gradually become a major concern of scholars. Dynamic capabilities theory has become a powerful perspective to analyze the new composition and new development direction of enterprise innovation capabilities in the digital context [16]. Digital innovation dynamic capability is an emerging concept proposed by academics to face the digital upgrading of organizational innovation capability in digital innovation context [17]. The improvement of digital innovation dynamic capabilities can help organizations to collect, store and analyze large amounts of data. Organizational-level digital innovation dynamic capabilities are divided into digital agility capabilities and restructuring innovation capabilities, in which digital agility capabilities follow the logic of opportunity, emphasizing the application of digital technology to perceive, capture, and respond to external opportunities, corresponding to the dynamic capabilities theory of perceiving opportunities and threats, and capturing opportunities; and restructuring innovation capabilities follow the logic of

value, emphasizing the re-integration and re-configuration of digital resources and traditional innovation resources to achieve value creation [18], corresponding to the reconfiguration of resources in dynamic capability theory. However, existing research lacks to explore the antecedents and impacts of digital innovation dynamic capabilities, especially whether digital innovation dynamic capabilities have an impact on firms' open innovation performance, which requires further research.

## 2.2 Open innovation

Open innovation promotes the construction of innovation networks between enterprises and external organizations, which enhances the ability of enterprises to access complementary resources and promotes the exchange and sharing of tacit and explicit knowledge by cooperating with externally affiliated partners. Meanwhile, reducing the risk of R&D activities by sharing the related costs to different channel partners is one of the main advantages of open innovation. Open innovation is based on the knowledge interaction view, which attaches importance to the flow of knowledge across organizational boundaries, interaction, integration and synergy among innovation elements [19].Under the open innovation paradigm, the innovation activities of enterprises are no longer controlled within the enterprise, and no longer emphasize the possession and control of innovation resources, but are embedded deeply in the network at the organizational level, and through cooperation with other various organizations, exchanging innovation resources, sharing innovation benefits, and ultimately gaining competitive advantages [20]. With the continuous progress and empowerment of the Internet and communication technology, innovation models continue to emerge and iteratively upgraded, presenting the following distinctive features: firstly, user-based open innovation, decentralization, information aggregation, focus on interaction and sharing as well as open open-source have become the main features of customer participation in innovation in the new era; secondly, platform-based open innovation, which changes the way of interaction between enterprises and the outside world by digitalization and networking, complementing each other with advantages and opening up to the outside world. The interaction between enterprises and the outside world in a digital and networked way has changed the way of interaction between enterprises and the outside world, complemented each other's strengths, and created value together, which provides a possible space for the development of the platform economy; finally, based on the ecosystem-based open innovation, the inter-firm competition has gone beyond the form of competitive game and market division [21], and the cross-border complementary fusion, empowerment of eco-partners, co-growth of the market, and the coexistence of coexisting and co-evolving have become its typical features. Existing research has confirmed that open innovation can break through the organizational boundary constraints and gain access to network heterogeneous resources and complementary capabilities that cannot be accessed by individual enterprises. However, current research is insufficient on the antecedents of open innovation performance of enterprises in the context of the digital economy, and further in-depth investigation is needed.

## 3. Review and theoretical hypotheses

### 3.1 Digitization transformation and open innovation performance

With the rapid development of digital technology, the conceptual definition of digital transformation is also being updated. Some studies suggest that digital transformation is the application of digital technologies such as data analytics [22], artificial intelligence, and other digital technologies to expand the scope of access to innovative knowledge through digital platforms, media, and other channels, thereby improving innovative partnerships and

innovation models; And some digital transformation as an innovative way of using digital technology as the main driving force to enable companies, partners and customers to co-create, communicate [23], deliver and maintain stakeholder relationships and create value. It can be seen that digital transformation is not only extending traditional innovation to online, but its key lies in better serving customers through in-depth data management, analysis and mining. Therefore, this paper argues that digital transformation is a new type of innovation that takes digital technology as the main driving force, builds up innovation channels through digital platforms, customizes marketing content using digital media, establishes precise and efficient connections with customers, and then realizes quantifiable, data-driven and intelligent innovation activities. Resource-based theory suggests that a firm's unique competitive advantage lies in its possession of scarce, valuable, inimitable, and irreplaceable resources [24]. The characteristics of digital transformation, such as platform diversity, target precision and deep interaction, are precisely the resources that enterprises need to acquire, and enterprises can effectively promote their open innovation performance by utilizing these characteristics of digital transformation.

Firstly, the platform diversity of digital transformation enhances the willingness of enterprises to open innovation performance. On the one hand, the vast number of digital transformation platforms provides enterprises with relatively scarce internal resources with richer access to external innovation resources. Utilizing these resources, enterprises can iteratively innovate more cost-effectively and improve the competitiveness of their products and services [25], which enables them to quickly respond to, guide, and anticipate customers' potential needs, and better deliver product value and stimulate customer consumption [26], which enhances their own willingness to open innovation performance; on the other hand, the digital platform diversity eliminates the two-dimensional limitations of geography and time, leading to the instantaneous generation of sales scenarios, giving rise to diversified market demand, and the resulting additional revenues stimulate competitors to follow up on the implementation of digital transformation strategies, prompting the platform competition to become increasingly transparent and intense, forcing companies to enhance their willingness to enhance their own open innovation performance in order to improve the conversion barriers and competitiveness of their products [27].

Secondly, the target precision of digital transformation improves the efficiency of enterprise innovation decision-making. On the one hand, enterprises adopting digital transformation are able to more accurately dig out the group characteristics of consumers and locate the target market through functions such as search weights and user profiles, realizing the targeted placement and allocation of innovation resources, thus effectively improving the accuracy of innovation decision-making; on the other hand, enterprises mastering massive amounts of data through digital transformation and relying on professional big data analysis technology and algorithms can more accurately assess the innovation incremental perception of products [28], accelerate the flexible innovation of product personalization, customization, and iteration, thus enhancing the efficiency of the enterprise's open innovation performance decision-making.

Thirdly, the deep interactivity of digital transformation strengthens firms' competitive advantage in innovation. On the one hand, firms have fewer ways to connect closely with their existing customers, and digital transformation improves firms' sensitivity to market segments, allowing them to obtain more timely feedback from customers on product improvements through social media and back-end communities [29], accelerating the interactive synergy between customer demand and firm innovation, and making it easier for firms to carry out iterative innovations that are adapted to market demand; On the other hand, based on social networks and CRM tools, digital transformation improves coordination among supply chains,

promotes market collaboration between enterprises and upstream suppliers based on demand, facilitates the emergence of symbiotic innovation ecosystems [30], stimulates cross-boundary cooperation and creative thinking in enterprise innovation, and is particularly conducive to accelerating the process of enterprise innovation, which significantly contributes to the emergence of a new generation of innovation. In particular, it is conducive to accelerating the innovation process of enterprises and significantly improving their innovation flexibility and interactivity, thus strengthening the competitive advantage of their open innovation performance.

In summary, digital transformation helps enterprises to improve their innovation competitive advantage, and enhances their innovation decision-making efficiency and willingness to open innovation performance, and promotes their open innovation performance. Based on the above analysis, this paper proposes the following hypotheses:

H1: Digital transformation positively affects open innovation performance.

## 3.2 The mediating role of digital innovation dynamic capability

Digital innovation dynamic capabilities are divided into digital agility capabilities and restructuring innovation capabilities, in which digital agility capabilities follow the logic of opportunity, emphasizing the application of digital technology to perceive, capture and respond to external opportunities, corresponding to perceiving opportunities and threats, and capturing opportunities in the theory of dynamic capabilities; and restructuring innovation capabilities follow the logic of value, emphasizing the reintegration and configuration of digital resources and traditional innovation resources to achieve value creation [31], corresponding to the reconfiguration of resources in dynamic capability theory. The characteristics of open innovation are mainly reflected in the organic integration of massive internal and external knowledge and the high risk brought about by facing uncertainty in various aspects such as technology realization and customer demand [32], which helps enterprises to utilize internal and external innovation resources to satisfy customer demand or explore emerging markets. And digital transformation has the potential to facilitate firms' open innovation performance by improving digital innovation dynamic capabilities (including digital agility capabilities and restructuring innovation capabilities).

Firstly, in environment-aware activities, digital agility capabilities support companies to quickly identify opportunities and threats in the market and generate new reference data in combination with previous business experience. By accurately predicting customer behavior and demand after generalization and comparison, innovative decisions are made that better fit the market, and uncertainty is transformed into controllable opportunities [33]. Not only that, in operational coordination activities, relying on the embedding of digital technology in the organizational management and operational structure, to promote the transformation and upgrading of business processes [34], to achieve the organization's openness and connectivity. Inside the organization, the adaptable and scalable open data platform composed of modular architecture realizes the efficient connection across levels and departments within the organization, guarantees that the developers can play a synergistic effect among different modules, and makes a substantial improvement in the innovation efficiency while guaranteeing the diversity of the innovation tasks [35]. Outside the organization, through the establishment of a digital platform to connect, including supply chain management, customer relationship management and other external systems, breaking the communication barriers with customers, more effective response to the diversified needs of customers, accurately and quickly to meet the market's dynamic needs, and promote the open innovation performance of the enterprise.

Secondly, the reorganization of innovation capability connects and combines internal and external digital technologies with digital resources, integrates material resources and non-material resources to break through the original industrial boundaries and realize the cross-field integration of creativity [36]. Enterprises can better integrate internal and external resources. On the one hand, enterprises can use adaptive reorganization activities to explore new uses of resources in the organization and stimulate synergies between different modules. On the other hand, organizations rely on adaptive reorganization activities to precipitate and integrate the wrong knowledge set accumulated in the active trial and error and the data generated by the internal business units of the organization, improve the accuracy of the organization's selection of resource objects for reorganization, and maximize the value of developing internal resources and data resources through resource patchwork [37]. Realize the dynamic matching between digital knowledge and digital resources. Open innovation activities can help enterprises search and find external complementary resources, further realize cross-regional and cross-industry value creation with multiple participation through interactive mechanisms such as sharing, and promote the network, synergy and ecology of resource allocation. Through external interaction mechanisms and related activities, it can significantly enhance the exploration ability of key elements such as external knowledge, thus increasing the width and depth of the internal knowledge pool of the organization, forming new knowledge that is difficult to imitate through integration and reorganization mechanisms, and promoting the development of enterprise open innovation.

To sum up, enterprises can better promote the development of enterprise open innovation by improving the dynamic capability of digital innovation. Based on the above analysis, this paper proposes the following research hypotheses:

H2: Digital innovation dynamic capability mediates the relationship between digital transformation and exploitative innovation.

### 3.3 The moderating role of environmental dynamics

In a high-environmental dynamism environment, where the risk of obsolescence of existing technologies and knowledge increases significantly, digital transformation reduces spatial constraints on resources by increasing the exploration of external digital resources, enhances connectivity between firms and other network members, improves heterogeneity of firms' access to knowledge and resources [38], and facilitates cooperation among innovation participants. In the low-technology dynamicity environment, innovation factor combinations and resource allocation modes tend to serve technologies on the mainstream track, with relatively few factors and resources able to participate in reorganization, and the reorganization of innovation capabilities lacks the object of use. In this context, enterprises tend to allocate innovation resources to the original technology field in order to maintain the leading position of the current technology and fully realize the maximization of the value of the existing technology [39]. At this time, the new technology generated is highly correlated with the original technology, the specialization of technology can be increased, and the demand for cross-industry restructuring of technology is weakened [40], which is not conducive to exerting the facilitating effect of restructuring innovation capability on open innovation.

In a highly dynamic technological environment, the emergence of emerging technology paths constantly challenges the dominant technology paths, and the original technology paths are no longer the only choices. Under this circumstance, enterprises can make full use of digital transformation to analyze the technology development trend in the turbulent and fragmented innovation intelligence [41], choose the breakthrough window according to their own technological innovation endowment, and improve the efficiency of open innovation performance through accurate judgment and concentration of resources. That is, high technological dynamics can provide sufficient space for digital

transformation to play, so as to better judge breakthrough opportunities [42]. In the low environmental dynamics environment, enterprises reduce the innovation amplitude and innovation frequency due to the interdependence of technology elements and high switching costs, and the technology development tends to maintain the status quo rather than actively get rid of the dilemma through the open innovation performance, so in the low-environmental dynamics environment, there is no need to increase the investment of enterprises to play the full function of the digital agility capability, and its promotion effect on the open innovation performance is not obvious.

In summary, environmental dynamism can positively moderate the relationship between digital transformation and open innovation performance. Based on the above research, this paper proposes the following hypotheses:

H3: Environmental dynamics have a moderating effect on the relationship between Digital transformation and open innovation performance.

The conceptual framework of this study is shown in Fig 1:

## 4. Methodology of the study

### 4.1 Sample selection and data sources

The research sample in this study consists of listed companies in the A-share market of China from 2008 to 2022. The following data treatments were applied in this study: (1) Financial industry companies were excluded due to significant differences in capital structure and accounting systems compared to other companies. (2) Companies marked with ST and * ST were excluded from the sample to avoid the influence of companies already in financial crisis or experiencing irreversible deterioration in their operational conditions. (3) Companies with severe missing data on key financial variables were excluded. (4) To mitigate the impact of outliers on the empirical model, all continuous variables were trimmed, with the top and bottom 1% of observations being discarded. The final sample included1862 companies with 19683 observations. The sample data were sourced from the CSMAR database, CNRDS database, and Wind database.

### 4.2 Variables definition

**4.2.1 Explained variables.** Open Innovation Performance (OIP). Drawing on the views of some scholars, this paper adopts the ratio of the number of patents applied by University-Industry-Research Cooperation (UIRC) to the number of patents applied by enterprises in

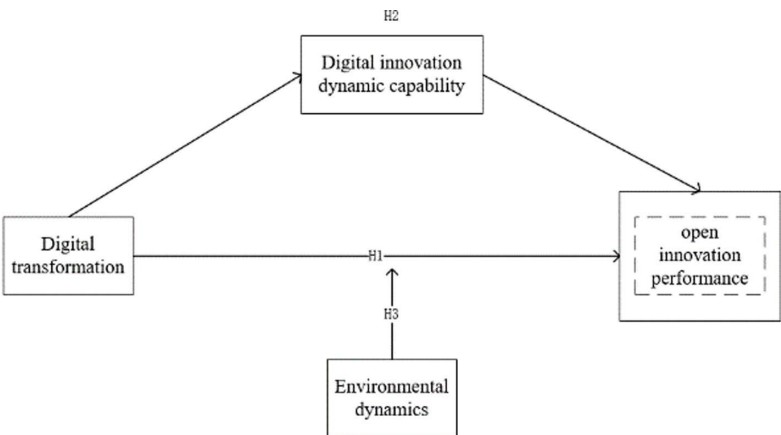

**Fig 1. Research framework diagram.**

a year [20], which is an objective indicator to represent the open innovation performance of enterprises. Among them, the number of patents applied by University-Industry-Research cooperation indicates the evaluation of the enterprise on the value of open innovation, the more patents applied by University-Industry-Research cooperation, the higher the evaluation of the enterprise on the open innovation, and it also reflects to a certain extent the degree of reliance on the external cooperation subject of the enterprise's technological innovation; the more patents applied by University-Industry-Research cooperation and the higher the ratio of the patents applied by cooperation, the higher the enterprise's performance in open innovation.

**4.2.2  Explanatory variables.**  Digital transformation (DT): In this study, I referred to the approach of Wu F [43] to construct a digital transformation index for listed companies based on the frequency of keywords related to digital technology in their annual reports. This study utilizes the methods of text analysis and word frequency statistics to construct an indicator system for measuring enterprise digital transformation. Text analysis can convert qualitative information into quantitative data, which is more widely used and more accurate in measuring indicators, and there is no lack of existing studies that use text analysis to mine keywords to construct indicators. The first step was to generate a database. I used Python web scraping techniques to collect annual reports of A-share listed companies from 2010 to 2021 on the China Securities Regulatory Commission website. These reports were then converted into txt format using a C++ program, which served as the database for subsequent keyword selection. The second step involved determining the keywords. Previous studies on digital transformation lacked a comprehensive selection of keywords, as they did not consider both the "fundamental digital technology level" and the "integration of digital technology with business operations level" aspects. Therefore, in this study, I defined the keywords for digital transformation from these two perspectives and supplemented the keyword library to provide a complete and comprehensive portrayal of the application of digital technology in companies. The final keyword corpus is presented in Fig 2. In the third step, the keyword frequency is calculated. Python is used in this study to retrieve, match, and calculate the frequency of keywords, resulting in a cumulative digital transformation index. Since the data exhibits a right-skewed distribution, logarithmic transformation is applied in this study.

**4.2.3  Mediating variables.**  Digital Innovation Dynamic Capability (DIDC): In this paper, based on Wang H [44], digital innovation dynamic capability is divided into digital agility capability (GAC) and reorganization innovation capability (RIC). Among them, digital agility capability follows the logic of opportunity, emphasizing the application of digital technology to perceive, capture and respond to external opportunities; and restructuring innovation capability follows the logic of value, emphasizing the reintegration and configuration of digital and traditional innovation resources to achieve value creation.

For digital agility capabilities, some studies suggest that digital resources have become an increasingly important strategic resource in corporate competition today [45], and companies need to continuously acquire and screen new valuable external digital resources in a timely manner, and this study draws on some research [25] to argue that it is the qualified digitally literate employees who can perform the digital resource acquisition and screening processes and carry digital knowledge. Digital agility is mainly reflected in the management of digital knowledge-based employees, which is a dynamic combination of digital knowledge, attitude and ability of digital knowledge-based employees, and it can better reflect the connotation that digital resources require digital knowledge-based employees to process, internalize and filter the introduced digital technology and knowledge through mental labor, and the number of digital R&D personnel in an enterprise can be used as a representative variable of digital

knowledge-based employees., and thus this paper chooses the annual number of digital R&D personnel, with logarithmic processing, as a measure of digital agility capability. For restructuring innovation capability, R&D investment is usually regarded as an important factor in determining a firm's restructuring innovation capability, therefore, this paper follows some research by scholars [46] and uses R&D investment as a proxy variable for restructuring



**Fig 2. Keyword corpus for digital transformation.**

innovation capability. Since different firms have different scales, the level of restructuring innovation capability is measured by the percentage of R&D expenses to operating income. Therefore, the digital innovation dynamic capability in this paper consists of two dimensions: digital agility capability and restructuring innovation capability, which can be calculated based on the standardization of each measurement index and the entropy weight method to determine its weight to get the level of the enterprise's digital innovation dynamic capability.

**4.2.4. Moderating variable.** Environmental dynamics (ED): Following the research method of Schneider [47], we measure the environmental dynamics (ED) of a company by using its performance volatility. To ensure more accurate measurement results, we first exclude the stable growth portion of the company's sales revenue and estimate the standard deviation of abnormal sales revenue for the sample companies over the past 5 years. This represents the company's environmental dynamics without industry adjustments. Next, we calculate the median environmental dynamics of the sample companies in the same industry for each year, which represents the industry's environmental dynamics. Finally, we measure the company's environmental dynamics by comparing its environmental dynamics without industry adjustments to the industry's environmental dynamics.

**4.2.5. Control variables.** Based on relevant literature, this study selects the following control variables: ① Scale: measured by the natural logarithm of total assets at the end of the period; ② Board: measured by the proportion of independent directors to the total number of board members; ③ Vrd: measured by the proportion of shares held by the largest shareholder of the company; ④ Leverage: the ratio of total liabilities to total assets; ⑤ Free cashflow: operating cash flow divided by total assets; ⑥ Grow: main business revenue growth rate. The measurement of variables is shown in Table 1:

## 4.3. Model design

To test the research hypothesis, this article establishes the following model:

$$OIP_{it} = \beta_0 + \beta_t DT_{it} + \sum \beta_k \text{Control}_{it} + \sum \text{Year} + \sum \text{Industry} + \varepsilon_{it} \qquad (1)$$

In the above model, "i" represents the company, "t" represents time. The dependent variable in the regression is open innovation performance (OIP), with digital transformation (DT) as the core explanatory variable. "Control" is the control variable, and "ε" represents the random error term in the model. To ensure more robust empirical results, the regression model in this study is treated in the following ways: Firstly, to control for the risk of

**Table 1. Measurement of variables.**

| Variable Type | Variable Name | Variable Symbols |
|---|---|---|
| Dependent variable | Open Innovation Performance | OIP |
| Independent variable | Digital transformation | DT |
| Mediating variables | Digital Innovation Dynamic Capability | DIDC |
| Moderating Variable | Environmental dynamics | ED |
| Control Variables | Scale | Scale |
| | Board | Board |
| | Vrd | Vrd |
| | Leverage | Leverage |
| | Free cashflow | Free cashflow |
| | Grow | Grow |

omitted variables, this study simultaneously controls for time fixed effects (Year) and industry fixed effects (Industry). Secondly, the t-statistics in all regression equations in this study are adjusted using robust standard errors clustered at the company level.

Second, in exploring the mediating effect of digital innovation dynamic capability, this study used a three-step approach to test the mediating effect of knowledge acquisition, as modeled by:

$$\text{Mediator} = \alpha_0 + \alpha_1 \text{DT}_{it} + \sum \alpha_k \text{Control}_{it} + \sum \text{Year} + \sum \text{Industry} + \varepsilon_{it} \qquad (2)$$

$$\text{OIP}_{it} = \beta_0' + \beta_1' \text{DT}_{it} + \beta_2' \text{Mediator} + \sum \beta_k' \text{Control}_{it} + \sum \text{Year} + \sum \text{Industry} + \varepsilon_{it} \qquad (3)$$

In this case, "Mediator" is the mediating variable of this study, representing digital innovation dynamic capability. The others variables are consistent with the baseline equation.

Third, the moderating effect of environmental dynamics between digital transformation and open innovation performance was explored. The specific model is as follows:

$$\text{OIP}_{it} = \alpha_3 + \beta_3 \text{DT}_{it} + \gamma \text{ED}_{it} + \eta \text{DT}_{it} \times \text{ED}_{i,t} + \sum \beta_{3k} \text{Control}_{it} + \sum \text{Year} + \sum \text{Industry} + \varepsilon_{it}$$
$$(4)$$

"ED" represents environmental dynamics for the businesses, which is the moderating variable in this study. The other variables are consistent with the baseline equation.

## 5. Research results

### 5.1 Descriptive statistics

Table 2 presents the descriptive statistics for the main variables. Descriptive statistics include the number of observations, mean, standard deviation, minimum value, and maximum value. For the variables of open innovation performance, the mean values are 0.383, with standard deviations of 0.127. The range of values is from 0 to 0.823. For the digital transformation of companies, the mean value is 1.630, with a range of values from 0 to 4.658 and a standard deviation of 1.218. The mean value for digital innovation dynamic capability is 0.316, with a noticeable difference between the maximum and minimum values of 0 and 0.861.

In this paper, Stata 16.0 software was applied to perform the test of Variance Inflation Factor (VIF) and to determine the presence of multicollinearity in these factors by calculating the VIF inflation factor between the variables. The results of the study show that the mean value

**Table 2. Descriptive statistics.**

| Variable | Obs | Mean | SD | Min | Max |
|---|---|---|---|---|---|
| OIP | 19683 | 0.383 | 0.127 | 0 | 0.823 |
| DT | 19683 | 1.630 | 1.218 | 0 | 4.658 |
| DIDC | 19683 | 0.316 | 0.527 | 0 | 0.861 |
| ED | 19683 | 0.052 | 0.027 | 0.015 | 0.079 |
| Scale | 19683 | 21.154 | 3.527 | 15.357 | 27.682 |
| Board | 19683 | 0.242 | 0.362 | 0 | 1 |
| Vrd | 19683 | 33.257 | 10.582 | 7.658 | 71.683 |
| Leverage | 19683 | 0.368 | 0.452 | 0.072 | 0.902 |
| Free cashflow | 19683 | 0.075 | 0.081 | −0.684 | 0.670 |
| Grow | 19683 | 0.153 | 0.126 | −0.528 | 1.973 |

of VIF is 2.73 and its maximum value is less than 10, which means that there is no serious problem of multicollinearity when making empirical estimation.

## 5.2. Main effect test

The results of the benchmark regression test are shown in Table 3. Model (1) controls for time and industry fixed effects, while Model (2) includes additional control variables. The results indicate that the regression coefficients for the level of digital transformation are significantly positive at the 1% level, suggesting that an increase in the level of digital transformation positively promotes open innovation performance in companies, providing support for the core hypothesis of this study.

## 5.3 Mediating effect test

The results of testing the mediating effect of the intermediary are shown in Table 4. Table 4 presents the results of testing the mediating effect of open innovation performance. Firstly, Models (1) in Table 4 serve as the baseline regressions for this study, consistent with previous findings, indicating that digital transformation significantly promotes open innovation performance. Models (2) in Table 4 reveal that the regression coefficients are significantly positive, indicating that digital transformation significantly and positively promotes firms' digital innovation dynamic capability. Models (3) in Table 4 represent the regression of open innovation performance on digital transformation and digital innovation dynamic capability. The coefficients for digital transformation are positive, as well as the coefficients for DIDC, and all pass the statistical significance test at the 1% level. Compared to Model (1), the coefficients for digital transformation in Model (3) decrease after including DIDC, but still pass the statistical significance test at the 1% level, indicating the presence of the mediating effect of digital innovation dynamic capability (DIDC). To ensure the robustness of the intermediate mechanisms, this study employs the Sobel test to examine the aforementioned mediation paths, and all pass the test.

**Table 3. Main effect test results.**

| Variables | (1) | (2) |
|---|---|---|
| | OIP | OIP |
| DT | 0.052** (0.005) | 0.071*** (0.004) |
| Scale | | 0.258* (0.064) |
| Board | | −0.003 (0.053) |
| Vrd | | 0.067 (0.068) |
| Leverage | | −0.135** (0.0627) |
| Free cashflow | | 0.147 (0.174) |
| Grow | | 0.052** (0.015) |
| Year | YES | YES |
| Industry | YES | YES |
| _cons | −0.063** (0.025) | −0.085** (0.011) |
| Obs | 19683 | 19683 |
| $R^2$ | 0.175 | 0.181 |

t-statistics in parentheses: * p < 0.1, ** p < 0.05, *** p < 0.01.

**Table 4. Mediating effect test results.**

| Variables | (1) | (2) | (3) |
|---|---|---|---|
| | **OIP** | **DIDC** | **OIP** |
| DT | 0.071*** (0.004) | 0.067*** (0.005) | 0.042*** (0.005) |
| DIDC | | | 0.039*** (0.003) |
| Controls | YES | YES | YES |
| Year | YES | YES | YES |
| Industry | YES | YES | YES |
| _cons | −0.057** (0.020) | −0.126* (0.061) | −0.169 (0.185) |
| Obs | 19683 | 19683 | 19683 |
| $R^2$ | 0.185 | 0.169 | 0.147 |

t-statistics in parentheses: * p < 0.1, ** p < 0.05, *** p < 0.01.

**Table 5. Moderating Effect test results.**

| Variables | (1) |
|---|---|
| | **OIP** |
| DT | 0.028*** (0.005) |
| ED | 0.052*** (0.015) |
| DT×ED | 0.137*** (0.024) |
| Controls | YES |
| Year | YES |
| Industry | YES |
| _cons | −0.136** (0.015) |
| Obs | 19683 |
| $R^2$ | 0.189 |

t-statistics in parentheses: * p < 0.1, ** p < 0.05, *** p < 0.01.

## 5.4 Moderating effect test

The results of the moderating effect test are presented in Table 5. Model (1) tested the moderating effect of environmental dynamics on the relationship between digital transformation and open innovation performance in companies. According to the test results, the interaction coefficient between environmental dynamics and digital transformation is 0.157, which is significant at the 1% confidence level and positive, indicating that the moderating effect of environmental dynamics is effective. Hypothesis 3 is validated. It can be seen that in situations of high environmental dynamics, companies are required to continuously absorb and create their own technological resources in response to changing demands, and strengthen their digital management capabilities, thereby creating favorable conditions for promoting intelligent transformation through digital innovation. Therefore, environmental dynamics can enhance the impact of digital transformation on open innovation performance.

## 6. Heterogeneity analysis

Although the above analysis reveals that digital transformation can positively impact company's open innovation performance, it is still unclear whether this impact exhibits heterogeneity in different contexts, further exploration is necessary. Existing literature provides valuable

insights for the heterogeneity analysis in this paper. Firstly, from the perspective of company nature attributes, there are differences between state-owned enterprises and non-state-owned enterprises in terms of resource base, governance structure, market competition environment, and many other aspects [48]. The digitalization motivation of enterprises is influenced by their nature, and compared with state-owned enterprises, non-state-owned enterprises are more motivated to introduce digital technology. Secondly, from the perspective of high-tech attributes, digital transformation is the forefront of technological innovation, and high-tech enterprises are more likely to invest corresponding resources to introduce digital technology and empower innovation activities. Based on the important evidence provided by the relevant literature mentioned above, this paper further examines the influence of company nature attributes, and high-tech attributes on the relationship between digital transformation and company's open innovation performance.

## 6.1  Property rights

This article distinguishes the samples into state-owned enterprise group and non-state-owned enterprise group based on the nature of the companies. The regression results are shown in Table 6. To compare the differences between the two groups, the coefficient of variation between groups is introduced, and the Bootstrap method is used to test the difference in coefficients between groups. The number of samples is set to 1000 times, and the final difference coefficient between groups is significant at the 1% significance level. By comparing the coefficient values of the two groups, it can be observed that the optimization effect of digital transformation is more pronounced in the non-state-owned enterprise group. This may be because the market competition among non-state-owned enterprises is usually more intense. To avoid being eliminated from the market due to insufficient innovation capabilities, non-state-owned enterprises are often more sensitive and attentive to changes in the external market environment and technological environment, and they need to utilize digital technology to promote research and development innovation activities.

## 6.2  High-tech characteristics

The study divided the samples into two groups, based on whether the company is a high-tech enterprise or not. Regression analysis was performed separately for each group, and the results are shown in Table 7. To compare the differences between the two groups, the coefficient of variation was introduced and a bootstrap method was used to test the differences in the coefficients between groups. The number of samples was set at 1000, and the coefficient differences between groups were found to be significant at a 1% level of significance. By comparing the

Table 6.  Test results based on different property rights.

| Variables | State-owned enterprise | Non-state-owned enterprise |
|---|---|---|
| DT | 0.036*** (0.005) | 0.025** (0.004) |
| Controls | YES | YES |
| Year | YES | YES |
| Industry | YES | YES |
| _cons | −0.582* (0.147) | −0.147 (0.164) |
| Obs | 7516 | 11354 |
| $R^2$ | 0.137 | 0.249 |

t-statistics in parentheses: * p < 0.1, ** p < 0.05, *** p < 0.01.

**Table 7. Test results based on different high-tech characteristic.**

| Variables | High-tech enterprise | Non- high-tech enterprise |
|---|---|---|
| DT | 0.041*** (0.005) | 0.029** (0.004) |
| Controls | YES | YES |
| Year | YES | YES |
| Industry | YES | YES |
| _cons | -0.320 (0.173) | −0.347* (0.109) |
| Obs | 6428 | 13062 |
| $R^2$ | 0.149 | 0.153 |

t-statistics in parentheses: * $p < 0.1$, ** $p < 0.05$, *** $p < 0.01$.

coefficient values between the two groups, it can be observed that the driving effect of digital transformation is more pronounced in the group of high-tech enterprises. This may be due to the strategic orientation and production management strategies of high-tech enterprises, which focus on technological innovation and have a high level of innovation activity. These enterprises attach great importance to technological innovation and have strong overall innovation capabilities. They are willing to invest a lot of resources in digital transformation and view it as a common understanding among high-tech companies.

## 7. Robustness test

### 7.1 Replacement of regression models

The robustness of the model was tested by replacing it with a higher-order joint fixed effects model. The model was examined using the higher-order joint fixed effects method, controlling for "Year × Industry". The regression results, as shown in Table 8, support the robustness of the conclusions made in this paper.

### 7.2 Endogeneity test

This article may have endogeneity issues due to sample selection and bidirectional causality. To address this, the instrumental variable method is used for testing and correction. Following the approach of Wang C [49], the average digital transformation of other companies within

**Table 8. Robustness test results.**

| Variables | (1) |
|---|---|
|  | OIP |
| DT | 0.035** (0.005) |
| Controls | YES |
| Year | NO |
| Industry | NO |
| Year × Industry | YES |
| _cons | −0.185 (0.171) |
| Obs | 19683 |
| $R^2$ | 0.192 |

t-statistics in parentheses: * $p < 0.1$, ** $p < 0.05$, *** $p < 0.01$.

**Table 9. Endogeneity test results.**

| Variables | (1) | (2) |
|---|---|---|
| | OIP | OIP |
| DT | | 0.057** |
| | | (0.005) |
| Other DT | 0.244*** | |
| | (0.004) | |
| Controls | YES | YES |
| Year | YES | YES |
| Industry | YES | YES |
| _cons | −0.168 (0.193) | −0.165 (0.185) |
| Obs | 15358 | 14273 |
| $R^2$ | 0.149 | 0.153 |

t-statistics in parentheses: * p < 0.1, ** p < 0.05, *** p < 0.01.

the same industry (Other DT) is used as the instrumental variable, and a two-stage instrumental variable method is applied for endogeneity handling. Table 9 presents the regression results. Model (1) shows the results of the first-stage regression, indicating that the instrumental variable Other DT has a significant positive impact on digital transformation, confirming the instrument's relevance. Furthermore, the instrumental variable passes the tests for overidentification (Anderson's canonical correlation statistic = 141.521, p < 0.01) and weak instrument (first-stage F statistic = 175.357). Model (2) presents the estimation results for the second stage, showing that the coefficients for digital transformation are all significantly positive, consistent with the empirical results mentioned earlier.

## 8. Conclusions and recommendations

Digital transformation has significant implications for helping businesses gain sustainable competitive advantages, driving technological advancements in industries, and even improving national competitiveness. This study examines the impact of digital transformation on open innovation performance in enterprises, as well as the mechanisms and moderating effects of external conditions, based on the data of A-share listed companies in the Shanghai and Shenzhen stock markets from 2008 to 2022. The research findings are as follows: (1) Digital transformation significantly promotes open innovation performance in enterprises, and this impact shows heterogeneity, with non-state-owned enterprises, and high-tech enterprises benefiting more from digital transformation in promoting open innovation performance. (2) Digital transformation can enhance the digital innovation dynamic capability of enterprises, thereby promoting open innovation performance. (3) The study incorporates important external conditions, and environmental dynamics plays a moderating role in the relationship between digital transformation and open innovation performance in enterprises.

Based on the conclusions of the above research, in order to promote digital transformation and upgrade open innovation performance in enterprises, this article puts forward the following policy suggestions from the perspectives of enterprises and government: Firstly, accelerate the implementation of the digital development model for enterprises. Enterprises should leverage the comparative advantages of new generation digital technologies in the process of digitization and informatization, create a favorable environment for digital innovation and operation, and further achieve comprehensive interconnection between people, machines, and objects. In the business logic driven by user value [50], enterprises

should explore and innovate in the core areas of digital technology, pay attention to the cultivation of digital talents, improve their own risk management system, and maximize the goal of open innovation performance. Secondly, deepen the digitization and digital industrialization of industries. Regarding industrial digitization, on the premise of ensuring data security and interoperability, the industry should strengthen resource integration and data sharing, and implement big data strategies within the industry. By optimizing and predicting data resources in the aggregation of the information network space, it promotes the upgrading of the digital transformation of industrial clusters. For digital industrialization, different industries such as IT industry, new energy industry, etc. should actively engage in cross-domain and cross-system digital businesses, build multi-party digital platforms for inter-industry interaction, and enhance the wide-ranging application and deep integration of cross-industry digitalization. Thirdly, create a favorable environment for digital technology infrastructure. The government should coordinate and promote the external environment, digital transformation, and strategic changes in a mutually reinforcing manner. While stabilizing the external market environment, it should also consider enterprise strategic changes and make strategic changes serve the digital economy, providing a fertile ground for the digital economy. It is necessary to quickly adapt to the pace of market digital transformation, create a favorable environment for digital development, realize digital participation among different departments, and promote the quality transformation, operational efficiency transformation, and driving capability transformation of the economic and social development.

## Supporting information

**S1 Data. Literature related data.**
(ZIP)

## Author contributions

**Writing – original draft:** Peiyao Qiu, Benrui Chang.

**Writing – review & editing:** Peiyao Qiu.

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
