## [Decision Letter · Decision Letter 0]

4 Apr 2024

PONE-D-23-29841The Impact of Digital transformation on enterprise dual innovation: the intermediary role of data-driven dynamic capabilitiesPLOS ONE

Dear Dr. Qiu,

Thank you for submitting your manuscript to PLOS ONE. After careful consideration, we feel that it has merit but does not fully meet PLOS ONE’s publication criteria as it currently stands. Therefore, we invite you to submit a revised version of the manuscript that addresses the points raised during the review process.

First of all I would like to offer** ** my sincere apologies for the delay in the review process of your paper. I understand the importance of timely feedback, and I regret any inconvenience caused by the prolonged review period. We encountered unexpected challenges in finding suitable reviewers who were available and willing to perform a thorough review. Despite our efforts, it took longer than anticipated to secure reviewers for your paper. I want to assure you that we take the peer review process seriously and strive to provide constructive feedback to authors. While one of the reviewers suggested accepting the paper, upon careful consideration, we found the comments made by the second reviewer to be insightful and valuable for improving the quality of your work. Therefore, I encourage you to carefully review the feedback provided by both reviewers and consider incorporating their suggestions into your paper. This collaborative approach will not only strengthen the overall quality of your research but also enhance its contribution to the academic community.

We look forward to receiving your revised manuscript.

Kind regards,

Jolanta Maj

Academic Editor

PLOS ONE

3. PLOS requires an ORCID iD for the corresponding author in Editorial Manager on papers submitted after December 6th, 2016. Please ensure that you have an ORCID iD and that it is validated in Editorial Manager. To do this, go to ‘Update my Information’ (in the upper left-hand corner of the main menu), and click on the Fetch/Validate link next to the ORCID field. This will take you to the ORCID site and allow you to create a new iD or authenticate a pre-existing iD in Editorial Manager. Please see the following video for instructions on linking an ORCID iD to your Editorial Manager account: https://www.youtube.com/watch?v=_xcclfuvtxQ".

5. We note that your Data Availability Statement is currently as follows: [All relevant data are within the manuscript and its Supporting Information files.]

Reviewers' comments:

Reviewer's Responses to Questions

**Comments to the Author**

1. Is the manuscript technically sound, and do the data support the conclusions?

Reviewer #1: Yes

Reviewer #2: Partly

2. Has the statistical analysis been performed appropriately and rigorously? 

Reviewer #1: Yes

Reviewer #2: N/A

3. Have the authors made all data underlying the findings in their manuscript fully available?

Reviewer #1: Yes

Reviewer #2: Yes

4. Is the manuscript presented in an intelligible fashion and written in standard English?

Reviewer #1: Yes

Reviewer #2: No

5. Review Comments to the Author

Reviewer #1: The article provides a comprehensive and insightful exploration of the impact of digital transformation on enterprise dual innovation. The authors employ a research paradigm centered on the "technology-economy" framework, drawing on the resource-based view, dynamic capability theory, and resource orchestration theory. This review aims to highlight the positive aspects of the study, acknowledging its valuable contributions to understanding the intricate dynamics of innovation in the digital era.

One of the key strengths of the article lies in its theoretical foundation, as it skillfully integrates the resource-based view, dynamic capability theory, and resource orchestration theory. By adopting this holistic approach, the study provides a robust framework for analyzing the multifaceted influence of digital transformation on enterprise dual innovation. This theoretical underpinning enhances the credibility of the research and contributes to filling the gap in previous studies that overlooked the data-driven dynamic characteristics of innovation development in the digital age.

The empirical testing of hypotheses using microdata from Chinese A-share listed companies spanning from 2010 to 2021 adds significant weight to the findings. The inclusion of real-world data lends practical relevance to the study, allowing for a nuanced understanding of how digital transformation positively promotes enterprise dual innovation. The acknowledgment of heterogeneity in the results further adds depth to the analysis, recognizing that the impact may vary across different contexts.

The identification of data-driven dynamic capabilities as a mediating factor between digital transformation and dual innovation is a noteworthy contribution. This finding underscores the importance of leveraging data in the innovation process, shedding light on the mechanisms through which digital transformation influences enterprise innovation dynamics. Additionally, the exploration of the moderating effect of environmental complexity enriches the study by revealing that the promotion effect of digital transformation on dual innovation is stronger in environments with higher complexity.

Reviewer #2: 1.The paper contains inconsistent citation formats, with some references using lowercase names while others use uppercase names. Please ensure that citation formats are unified according to the journal's requirements.

2.Please provide an explanation regarding the presence of multicollinearity issues in the analysis.

3.Please elucidate where the authors have made theoretical contributions in comparison to existing research.

4.In the Theoretical Hypotheses section, the hypotheses proposed by the authors lack sufficient literature support. Please supplement with relevant references to strengthen the theoretical hypotheses.

5.In “4.2.3 Mediating variables”, the authors measure data-driven dynamic capabilities using data related to R&D. Please explain why general R&D data can capture a firm's data-driven dynamic capabilities rather than R&D specifically related to digitalization. Additionally, I was unable to locate the reference to Wu et al. (2021). Please verify and provide the correct citation details.

6.Please explain the rationale behind using textual analysis to measure digital transformation. I am skeptical about the authors' use of the MD&A section of annual reports to capture firm actions, as executive teams may employ rhetoric rather than actual actions in annual reports.

6. PLOS authors have the option to publish the peer review history of their article (what does this mean? ). If published, this will include your full peer review and any attached files.

**Do you want your identity to be public for this peer review?** For information about this choice, including consent withdrawal, please see our Privacy Policy .

Reviewer #1: No

Reviewer #2: No

---

## [Author Response · Author response to Decision Letter 1]

19 Sep 2024

Dear Editor：

Your last reply letter made some suggestions, all of which were very professional. I have made changes according to these suggestions line by line.

In response to reviewer's suggestions, I've made the following changes:

(1) Response 1: This paper has been revised to change the citation format to be consistent based on the reviewer's comments. The citation format is standardized according to the journal's requirements.

(2) Response 2: The multicollinearity problem in the article has been analyzed in this paper.

(3) Response 3: The paper has been revised in response to the reviewers' comments, indicating where the paper makes a theoretical contribution compared to existing research.

(4) Response 4: This paper has been revised based on the comments of the reviewers, and in the theoretical assumptions section, relevant references have been added to strengthen the theoretical assumptions.

(5) Response 5: This paper has been modified in response to comments, and in “4.2.3 Mediating Variables”, the authors measure mediating variables and use digitization-related measurement variables. In addition, a citation was added for Wu F et al. (2021), verified and provided the correct citation information.

Wu, F., Hu, H., Lin, H. & Ren, X. Corporate digital transformation and capital market performance-Empirical evidence from stock liquidity. Manage. World. 2021;37: 130–144 .

(6) Response 6: This article, revised in response to commenters, explains why text analytics are used to measure the rationale behind digital transformation. I have deleted content from the MD&A section of the article that uses annual reports.

Yours sincerely

Qiu Peiyao

407853679@qq.com

---

## [Decision Letter · Decision Letter 1]

22 Nov 2024

PONE-D-23-29841R1The impact of digital transformation on open innovation performance: the intermediary role of digital innovation dynamic capabilityPLOS ONE

Dear Dr. Qiu,

Thank you for submitting your manuscript to PLOS ONE. After careful consideration, we feel that it has merit but does not fully meet PLOS ONE’s publication criteria as it currently stands. Therefore, we invite you to submit a revised version of the manuscript that addresses the points raised during the review process.

**ACADEMIC EDITOR: ** Overall, the authors have made significant efforts to revise this manuscript and have addressed many of the concerns I previously raised. At this stage, I have only one remaining question.

The authors mention in the introduction, “However, previous studies have... less explored the impact mechanism of digital transformation on enterprise open innovation performance from the perspective of digital innovation dynamic capabilities.” To strengthen this argument, the authors should specify the perspectives from which existing research has explored the relationship between digital transformation and open innovation performance, underscoring the limitations of these studies.**.**  

We look forward to receiving your revised manuscript.

Kind regards,

Reza Rostamzadeh

Academic Editor

PLOS ONE

Journal Requirements:

Reviewers' comments:

Reviewer's Responses to Questions

**Comments to the Author**

1. If the authors have adequately addressed your comments raised in a previous round of review and you feel that this manuscript is now acceptable for publication, you may indicate that here to bypass the “Comments to the Author” section, enter your conflict of interest statement in the “Confidential to Editor” section, and submit your "Accept" recommendation.

Reviewer #2: (No Response)

2. Is the manuscript technically sound, and do the data support the conclusions?

Reviewer #2: Yes

3. Has the statistical analysis been performed appropriately and rigorously? 

Reviewer #2: Yes

4. Have the authors made all data underlying the findings in their manuscript fully available?

Reviewer #2: Yes

5. Is the manuscript presented in an intelligible fashion and written in standard English?

Reviewer #2: Yes

6. Review Comments to the Author

Reviewer #2: Overall, the authors have made significant efforts to revise this manuscript and have addressed many of the concerns I previously raised. At this stage, I have only one remaining question.

The authors mention in the introduction, “However, previous studies have... less explored the impact mechanism of digital transformation on enterprise open innovation performance from the perspective of digital innovation dynamic capabilities.” To strengthen this argument, the authors should specify the perspectives from which existing research has explored the relationship between digital transformation and open innovation performance, underscoring the limitations of these studies.

7. PLOS authors have the option to publish the peer review history of their article (what does this mean? ). If published, this will include your full peer review and any attached files.

**Do you want your identity to be public for this peer review?** For information about this choice, including consent withdrawal, please see our Privacy Policy .

Reviewer #2: No

---

## [Author Response · Author response to Decision Letter 2]

30 Dec 2024

In response to reviewer's suggestions, I've made the following changes:

(1) Response 1: This article has been revised to reflect the commenters' suggestions, which reinforce this argument: "Previous studies have less explored the mechanism of digital transformation's impact on firms' open innovation performance from the perspective of digital innovation dynamic capability." This paper describes the existing research on the impact of digital transformation on open innovation performance and its mechanism from the perspective of knowledge management, enterprise digital process, etc., and explains the limitations of these studies and the value of this research.

---

## [Editor Report · Decision Letter 2]

5 Jan 2025

The impact of digital transformation on open innovation performance: the intermediary role of digital innovation dynamic capability

PONE-D-23-29841R2

Dear Dr. Qiu,

We’re pleased to inform you that your manuscript has been judged scientifically suitable for publication and will be formally accepted for publication once it meets all outstanding technical requirements.

Kind regards,

Reza Rostamzadeh

Academic Editor

PLOS ONE
---

## [Editor Report · Acceptance letter]

PONE-D-23-29841R2

PLOS ONE

Dear Dr. Qiu,

I'm pleased to inform you that your manuscript has been deemed suitable for publication in PLOS ONE. Congratulations! Your manuscript is now being handed over to our production team.

Kind regards,

on behalf of

Dr. Reza Rostamzadeh

Academic Editor

PLOS ONE